# Left Ventricular Assist Devices: A Primer for the Non-Mechanical Circulatory Support Provider

**DOI:** 10.3390/jcm11092575

**Published:** 2022-05-04

**Authors:** Gregory S. Troutman, Michael V. Genuardi

**Affiliations:** 1Department of Medicine, Hospital of the University of Pennsylvania, Philadelphia, PA 19104, USA; tgregory@pennmedicine.upenn.edu; 2Perelman School of Medicine, University of Pennsylvania, Philadelphia, PA 19104, USA

**Keywords:** left ventricular assist devices, heart failure with reduced ejection fraction, right heart failure

## Abstract

Survival after implant of a left ventricular assist device (LVAD) continues to improve for patients with end-stage heart failure. Meanwhile, more patients are implanted with a destination therapy, rather than bridge-to-transplant, indication, meaning the population of patients living long-term on LVADs will continue to grow. Non-LVAD healthcare providers will encounter such patients in their scope of practice, and familiarity and comfort with the physiology and operation of these devices and common problems is essential. This review article describes the history, development, and operation of the modern LVAD. Common LVAD-related complications such as bleeding, infection, stroke, and right heart failure are reviewed and an approach to the patient with an LVAD is suggested. Nominal operating parameters and device response to various physiologic conditions, including hypo- and hypervolemia, hypertension, and device failure, are reviewed.

## 1. Introduction

The prevalence of heart failure (HF) continues to rise in the U.S. as the population ages. An estimated 6 million American adults have HF according to 2015–2018 data [1]. Pharmacologic and device-based treatment advances have helped to extend the lifespan and slow disease progression of HF, leading to a growing population that the European Society for Cardiology describes as “advanced, chronic HF” [2].

After the careful exclusion and treatment of reversible causes, modern care of the patient with HF with reduced ejection fraction (HFrEF) is focused on rapid initiation and uptitration of guideline-directed medical therapy with contemporary “quadruple medical therapy”, as follows: angiotensin receptor-neprilysin inhibitors, evidence-based beta-blockers, mineralocorticoid receptor antagonists, and sodium glucose cotransporter 2 inhibitors [3]. Aggressive implementation of quadruple therapy can lead to recovery and significant reduction of morbidity/mortality associated with HFrEF. Device therapies other than mechanical circulatory support (MCS), such as cardiac resynchronization therapy, have also lessened disease morbidity and mortality [4]. However, many patients will still develop progressive disease despite the promise of these therapies and may require more advanced measures.

Heart failure ranges in severity, with the American Heart Association defining the most severe as Stage D HF, designating “the patient with end-stage disease who requires specialized treatment strategies such as mechanical circulatory support, continuous inotropic infusions, cardiac transplantation, or hospice care” [5]. Left ventricular assist device (LVAD) therapy was initially developed as short-term MCS for patients with end-stage HFrEF as a bridge-to-transplant (BTT), but advances in the durability and portability of LVADs over the past decades have led to a revolution in the typical LVAD patient profile. Now, most patients receiving an LVAD implant are destination therapy (DT), rather than BTT [1,6]. Other device indications include bridge-to-recovery in patients who are hoped to improve enough for explant, and bridge-to-candidacy (or the essentially interchangeable “bridge-to-decision” or “DT-modifiable”), which intends to capture patients who are not transplant candidates at the time of implant but have reversible contraindications. Several research consortia, including the influential Interagency Registry for Mechanically Assisted Circulatory Support (Intermacs), as well as many LVAD clinicians colloquially, use these terms to supplement DT and BTT where appropriate. Importantly, the Centers for Medicare & Medicaid Services, which sets standards for public insurance programs in the U.S., had previously used only the DT and BTT labels. As of 2020, however, all reference to DT and BTT were removed from guidelines in an acknowledgement of the strong evidence base for LVAD therapy irrespective of implant indication [7].

Between 2010–2019, over 25,000 patients underwent implantation of a durable, continuous-flow LVAD, with nearly 3200 in 2019 [8]. The increasing proportion of annual DT implants and the greater survival of patients on long-term MCS has created a growing population of long-term LVAD patients presenting for care for myriad LVAD and non-LVAD related health conditions. This review is meant to serve as a primer for physicians and other care providers who encounter LVAD patients in their regular scope of practice, although they are not primarily advanced heart failure or MCS practitioners.

## 2. The LVAD Patient Profile

Determining optimal timing for LVAD referral can be challenging for heart failure patients but is critical, as late referral is associated with poorer outcomes after LVAD implantation. In particular, secondary end-organ dysfunction often precludes LVAD implantation in patients who are not promptly referred or evaluated [9]. The phrase “I NEED HELP” is useful as a memory aide for the core factors defining the progression from stable to advanced heart failure. [10]. The mnemonic stands for the following: Inotropes (history of or current need), NYHA class (III or IV symptoms), End-organ dysfunction (particularly renal or hepatic dysfunction), Ejection fraction (<35%), Defibrillator shocks (recurrent for ventricular tachycardia or fibrillation), Recurrent hospitalization (>1 in the past year), Edema/Escalating diuretic dosing (implying diuretic refractoriness), Low blood pressure (systolic < 90 mmHg), or Prognostic medications (intolerance of neurohormonal blockage). All of these “warning signs” attempt to identify patients with advanced and/or progressive HFrEF despite optimal medical therapy who may be considered for LVAD implantation. The challenge remains to target these patients earlier in their advanced HF course, i.e., prior to the development of shock, renal or hepatic dysfunction, or hospital dependence [11]. Delayed referral and LVAD implantation for patients with progressive decline on inotropic support or those in critical cardiogenic shock leads to increased complications and mortality after LVAD implantation [12].

It is important to note that, despite advances in medical and mechanical therapies, racial and ethnic disparities in heart failure persist across incidence, hospitalization, and outcomes. Over the past two decades, disparities in heart failure mortality in African Americans have worsened [13,14]. Racial and ethnic disparities in LVAD utilization have narrowed, however disparities in transplantation and outcomes remain [15,16]. Individual, organizational, and structural interventions are necessary to combat the structural racism that exists in heart failures therapies and our medical system as a whole.

## 3. Toward the Modern Continuous Flow Era

It has been almost six decades since the first successful ventricular assist device was used in a human patient after years of unsuccessful attempts to develop a practical total artificial heart [17]. The first LVADs were primarily used for post-cardiotomy shock in what would now be termed a “bridge-to-recovery” indication. Simple in construction and external to the patient, they consisted of a hemi-spherical metallic casing containing an inflatable bladder, rhythmically inflated with CO_2_. The cyclical negative and positive pressure drew ventricular blood into the hemi-spherical housing before ejecting it into the target blood vessel via surgically placed conduits. The device was timed via electrocardiographic synchronization to pump during diastole, when the aortic valve was closed, or else timing could be determined manually. Level of support (flow rate) was regulated by changing the ratio of native cardiac cycles to LVAD inflations (analogous to the support ratio on a modern intra-aortic balloon pump) or by changing the bladder inflation volume. One-hour battery powered excursions were possible.

The field of ventricular assist developed in parallel to cardiac transplantation, the latter first performed in the late 1960s. Various devices were employed successfully for bridge-to-recovery or bridge-to-transplant throughout the 1980s and 1990s [18]. However, the most successful LVADs for intermediate and long-term support were pulsatile, retaining close design similarity to the first devices used decades prior, such as the HeartMate IP and later VE/XVE models and the Novacor LVAS [19]. The HeartMate VE/XVE system from Thoratec (Pleasanton, CA, USA) was a notable breakthrough in LVAD history, obtaining the first approval for use as long-term support in transplant-ineligible patients, for so called destination therapy. This approval was gained on the basis of 2001’s REMATCH trial, showing superior mortality among end-stage heart failure patients implanted with the fully implantable, battery-powered HeartMate VE as compared to medical therapy alone [6]. More recently, concerns over mechanical reliability and hemocompatibility have ushered in modern continuous flow pumps. The continuous flow long-term support era was launched by the approval of the Thoratec HeartMate II axial flow device, based on clear improvements in device reliability and survival, in 2009 [20]. Specifically, overall 2-year survival free of stroke or device malfunction rose dramatically, from 11 to 46%, with continuous axial flow as compared to pulsatile flow. Overall mortality was reduced by over 40%. Early fears about nonphysiologic continuous flow on organs such as the gut and brain did not materialize in a clinically significant way [21]. With the era of highly reliable continuous flow pumps well established, the field shifted yet again when two new devices miniaturized the pump rotor and moved the rotor housing from the abdomen to intra-pericardial, directly adjacent to the LV apex. These devices, the Heartware VAD (“HVAD”, Medtronic, Minneapolis, MN, USA) and the HeartMate 3 (now owned by Abbott of Chicago, Chicago, IL, USA), had further reliability advantages [22,23]. The HVAD was withdrawn from the market in 2021 over concerns about increased risk of stroke [24], leaving the HeartMate 3 system the only long-term ventricular assist device in North America as of 2022.

## 4. The Left Ventricular Assist Device Components

The main components of the current era, continuous flow, LVADs include the LVAD inflow cannula, pump, and outflow cannula, an electric controller, batteries and a percutaneous driveline (Figure 1). The percutaneous driveline carries electrical cables and an air vent to the battery packs worn on a shoulder holster and electric controls worn on a belt [25]. The inflow cannula is inserted into the apex of the left ventricle (LV) with the outflow cannula delivering blood to the ascending aorta (Figure 2). Blood returns from the lungs through left-sided native anatomy to the point of the LV, and exits via the LV apex across an inflow valve into the prosthetic pump and then the ascending aorta. There is typically some amount of competitive native flow across the aortic valve during systole, however in many cases the LVAD provides “full support” and the aortic valve does not open.

## 5. Clinical Scenarios and LVAD Emergencies

### 5.1. Hemodynamic Basics

The hemodynamics and physiology of contemporary LVADs are unique. These continuous flow, valveless pumps generate blood flow that is inversely proportional to the pressure gradient between the aortic and LV pressure. In systole, the flow through the pumps is increased, and it is decreased during diastole. The difference between peak systolic and trough diastolic flow is termed the device “pulsatility” (“PI” on Abbott/Thoratec devices) and may or may not correspond to a measurable pulse pressure in the traditional sense. Because LV contractility is impaired in the typical LVAD patient, diminished pulse pressure is the norm [26]

Both centrifugal devices, the HeartMate 3 and the HVAD, have features that briefly vary the device speed (rpm) in periodic fashion to accomplish several physiologic and fluid dynamic goals. The HVAD’s “Lavare cycle” is turned on and off via the device settings as an option, while the HeartMate 3’s “artificial pulse” feature is on by default and cannot be disabled. The artificial pulse is designed to very briefly (0.15 s) drop the device speed by 2000 rpm below the set speed, then briefly (0.2 s) increase to 2000 rpm above the set speed, then return to the set speed. The cycle repeats every 2 s and accounts for the musical pitch variation heard on stethoscope auscultation at a 30 Hz frequency. While the HeartMate 3 artificial pulse cycling will often cause small (<15 mmHg) variations on invasively monitored blood pressure or spectral Doppler echocardiography (Figure 3), it is not typically palpable on physical examination. The HVAD’s Lavare cycle is similar in concept but has notable differences: the speed changes occur once every 60 s (as opposed to every 2 s) and last for longer (2 and 1 s).

LVADs are both extremely preload dependent and afterload sensitive. This leads to important implications for inpatient management and perioperative care.

### 5.2. Peri-Procedural Management

Anticoagulation with an oral vitamin K antagonist (e.g., warfarin) is recommended for all LVAD-supported patients [27]. Most LVAD patients are also maintained on an antiplatelet, typically 81–325 mg/day of aspirin. Whether the anti-platelet can be safely left off the regimen in HeartMate 3 patients is the subject of ARIES, an ongoing randomized controlled trial [28]. Full dose anticoagulation should be continued during operative procedures as able, as is in other cardiac surgeries [29]. Bridging anticoagulation with IV unfractionated heparin is almost always used in cases where operation on anticoagulation is not feasible. Given the preload dependence and afterload sensitivity, volume status optimization and maintenance of a mean arterial pressure (MAP) of 70–90 mmHg are crucial for preserved end-organ function in the pre-, peri-, and post-procedural period. Approximately 30% of all LVAD-supported patients require non-cardiac surgery, most frequently gastrointestinal endoscopy, invasive cardiac catheterization or electrophysiological interventions [30]. Most non-cardiac surgeries can be safety performed in LVAD patients with interruption of anticoagulation and guidance from experienced LVAD teams.

### 5.3. Acute Bleeding

Patients with LVADs are at high risk for complications of bleeding. This is due to obligate anticoagulation and hemocompatibility—i.e., the blood-device interface between the LVAD pump and blood itself [31]. Acquired von Willebrand syndrome, previously described in patients with severe aortic stenosis or severe heart failure, can be seen in LVAD-supported patients. This is thought to be due to degradation of von Willebrand Factor (vWF), the large multimeric glycoprotein that promotes platelet aggregation and hemostasis at sites of vascular injury. Almost all LVAD patients exhibit loss of vWF activity [32], which resolves after LVAD explant or transplant [33]

Bleeding can occur in up to 22% of LVAD-supported patients [34]. In the case of acute blood loss anemia, early management of LVAD patients mirrors that of non-LVAD patients. For severe gastrointestinal (GI) bleeding with hemodynamic compromise, MAP of 60–80 mmHg should be maintained, along with typical GI bleed management such as maintaining adequate intravascular access and IV proton pump inhibitor therapy. Reversal of the vitamin K antagonist may be considered in cases of blood loss causing hemodynamic compromise and supratherapeutic international normalized ratio (INR).

In addition to GI bleeding, intracranial/subarachnoid hemorrhage are feared complications [20,35]. Treatment decisions for hemorrhagic stroke are guided by whether the patient had a primary hemorrhagic stroke (intracerebral hemorrhage or subarachnoid hemorrhage) versus hemorrhagic conversion of an ischemic stroke. Multidisciplinary consultation is recommended, given the need for complex decision-making regarding reversal of anticoagulation, risks/benefits of systemic lysis, and blood pressure management in the setting of acute stroke. The risk of stroke was notably elevated in the now recalled HVAD, especially in patients with MAP > 90 mmHg [36]. Meanwhile, some evidence suggests that chronic MAP < 75 mmHg is also associated with stroke and all-cause mortality in a population comprising multiple LVAD models [37]. An appropriate MAP target specific to the HeartMate 3 device that may prevent neurologic events, bleeding, and aortic insufficiency is unknown and under active investigation.

### 5.4. Thrombosis

Pump thrombosis, though less common than with earlier generation devices, still may occur in anywhere from 2–13% of adult LVAD-supported patients [22,38]. Thrombosis can occur in the inflow cannula which siphons blood from the LV apex, intra-device/pump rotor itself, or along the outflow cannula that feeds into the ascending aorta. It is possible for thrombi to be created within the LVAD, or to form in the native left atrium or LV and travel into the pump housing.

Inflow cannula and intra-pump thrombosis are suspected in patients with worsening heart failure symptoms, particularly signs of left-sided heart failure such as pulmonary edema. Ingestion of clots or formation of a clot on the pump rotor will lead to increased power consumption and so-called “power spikes” (Figure 4). Importantly, the LVAD flow displayed on the controller is imputed from power consumption. While rotor thrombosis will cause increased power consumption (it requires more power to spin the clot-laden rotor), actual pump flow will *decrease*, despite the controller showing a flow *increase* (Table 1).

Pump thrombosis in the rotor itself should be suspected when patients present with symptoms or laboratory findings of hemolysis, such as lactate dehydrogenase elevation or tea-colored urine. Occasionally, echocardiographic “ramp studies” may be helpful to diagnose thrombosis. In a ramp study, LV parameters such as size, aortic valve opening, mitral regurgitation, and inflow cannula turbulence are observed while varying pump speed [39]. A malfunctioning/thrombosed device will lead to only trivial changes in LV unloading, despite changes to pump speed.

Lastly, thrombosis can occur in the outflow graft, here both due to “internal” clot formation within the outflow graft itself and external compression from pressure on this conduit. The rare but serious complication of outflow graft occlusion has been reported in the HeartMate 3 due to extrinsic impingement [40] and twist in the outflow graft [41], the latter addressed with a technical advisory and change in surgical technique in LVADs implanted since 2019.

First line therapy for device thrombosis at any location is emergent transport to a well-equipped LVAD center and systemic anticoagulation. Continued thrombosis, or thrombosis with heart failure symptoms or hemodynamic compromise, is generally treated with systemic lysis after careful exclusion for a source of occult hemorrhage, especially intra-cranial. Some cases require device exchange, which can be a morbid procedure and is reserved only for refractory cases [42]. In patients who are BTT or are transplantable, urgent transplant is a viable option. In the U.S., the United Network for Organ Sharing waitlist criteria explicitly allows substantial escalation of waitlist priority for LVAD patients with clinically significant thrombosis.

### 5.5. Infections

The driveline exit site in the abdominal wall is a common source of infection in LVAD patients, especially in patients with certain comorbidities common to the advanced HF population, such as obesity or diabetes. The ISHLT Infectious Disease Working Group [43] has differentiated infections in LVAD patients to the following three categories: LVAD-specific infections, LVAD-related infections, and non-LVAD related infections. LVAD-specific infections include infections related directly to LVAD hardware and can often be difficult to diagnose and eradicate due to biofilm formation. Thorough infectious work-up, initiation of broad antimicrobial treatment, and engagement of multidisciplinary teams is crucial for treatment of suspected infections in LVAD-supported patients. Workup of suspected driveline infection should include photography and sterile culture of the driveline exit site, blood cultures, abdominal well ultrasonography and/or chest/abdomen CT imaging, and empiric antibiotics as indicated.

### 5.6. Right Ventricular Failure

Right ventricular (RV) failure is a common complication after LVAD implantation and is associated with increased morbidity and mortality [44,45]. Prediction of RV failure after LVAD can be challenging [46], and right heart failure is a graded spectrum, with severity ranging from subtle findings to prolonged inotropic support or mechanical right ventricular support. The spectrum of severity aligns both with mortality [47] and functional capacity [48] after LVAD implantation. As such, it is critical to identify patients at high risk of post-operative RV failure, in order to select patients who might benefit from immediate temporary right ventricular support device or who might not be candidates for LVAD implantation at all.

Diagnosis of RV failure is similar to diagnosis of HF in the general population, except that symptoms will be right-sided. For example, fatigue, edema, ascites, liver and kidney injury, elevated jugular venous pressure, and GI complaints due to gut edema or low cardiac output will be typically observed, whereas pulmonary edema and related symptoms such as dyspnea, orthopnea, and paroxysmal nocturnal dyspnea will not be featured in the presentation.

### 5.7. Hypertension

Hypertension, typically defined as MAP > 90 mmHg, Ref. [27] is common among LVAD patients and is associated with ischemic stroke, intracranial hemorrhage, pump thrombosis, aortic regurgitation, and ventricular arrhythmias [49,50]. In addition to systemic complications of elevated blood pressure, increased afterload will decrease LVAD pump flow, leading to the typical combination of low flow alarms with high pulsatility.

As discussed above, typical LVAD patient pulse pressures are less than 15 mmHg and are non-palpable, nor detectable by an automated (oscillometric) blood pressure cuff. Arterial line direct measurement is the most accurate and reliable method for blood pressure monitoring of LVAD patients, but not always necessary or feasible. For the majority of LVAD patients, a vascular Doppler transducer placed over the brachial artery combined with manual sphygmomanometer can be used to measure a mean arterial pressure. Some caution should be used in hypertensive or highly pulsatile patients, as the Doppler pressure might more closely reflect a systolic pressure, rather than a MAP [51]. A combination of blood pressure measurement techniques including Doppler, manual auscultatory, and oscillometric can be used when there is doubt.

Renin-angiotensin-aldosterone system blockage with angiotensin-converting enzymes inhibitors, angiotensin II receptor blockers, or mineralocorticoid receptor antagonists are helpful agents, both for blood pressure control and prevention of continued adverse remodeling in HFrEF patients [29]. Carvedilol is also helpful for mixed alpha/beta adrenergic antagonism, although caution is needed in patients with underlying RV dysfunction or failure with the use of negative inotropy that comes with beta-blockade.

### 5.8. The Non-Responsive LVAD Patient

Patients with LVADs often lack a palpable pulse, and measurement of peripheral oxygen saturation may be challenging with standard equipment, making evaluation of perfusion in the non-responsive patient occasionally challenging for an inexperienced provider. Apnea, agonal breathing, unresponsiveness, or pallor should all raise urgent concern for lack of effective perfusion. Capnography can be particularly helpful to both assess unresponsive patients and guide resuscitation efforts. Although chest compressions for the pulseless LVAD patient may be intuitive, there are small retrospective reports [52]. that show a delay in cardiopulmonary resuscitation efforts for LVAD patients compared to non-LVAD patients. This could be due to concern of causing pump dislodgement or other mechanical complications as well as the difficulty in rapidly assessing perfusion. Despite these fears, current recommendations are for standard cardiopulmonary resuscitation in the case of pulselessness/ill-perfusion [53].

## 6. Putting It Together: Approach to the LVAD Patient

Thoughtful evaluation of a patient with an LVAD implant takes into account both potential device-related and non-device-related problems. After an initial survey assessing perfusion, a patient history and physical examination should be undertaken, which focuses on both the patient’s chief complaint and also common LVAD-specific issues. Specifically, careful attention should be paid to subtle signs of anemia or hypovolemia, which may suggest GI bleeding. Stool and urine quality and quantity should always be addressed as changes might suggest bleeding, hemolysis, or heart failure. Signs of left-sided heart failure, such as dyspnea or orthopnea, should always prompt consideration of suboptimal device performance, such as thrombosis or obstruction or hypertension. LVAD patients often have impaired functional immunity and risk factors that predispose to infection such as healthcare contact and diabetes. They are at high risk of serious complication from infection including, from 2019 novel coronavirus disease [54]. Careful assessment of driveline/device and non-driveline/device infection is critical.

One of the more common reasons to present to medical care is the “low-flow” alarm. The HVAD device will alarm at flows < 1.0 L/min by default, and is typically set to alarm at <2.0 L/min. The HeartMate 3 will alarm at flow < 2.5 L/min by default. On the HVAD, low-flow is a medium priority alarm, and is signified by an audible alert and a flashing yellow light on the patient’s controller. The HVAD patient controller displays only active alarms; review of recent acknowledged/cleared alarms requires connection to the bedside controller console by a healthcare provider. The HeartMate 3 will allow scrolling through recent alarms by pressing the display button (a round button with a square icon).

The HVAD typical operating range is 2400–3200 rpm. The device typically draws 2.5–8.5 Watts, with higher power draws at faster set speeds, and will flow at 3–8 L/min, again depending on the set speed, hematocrit, and loading conditions. The “pulsatility,” or the variation in flow between the peak flow in systole and trough flow in diastole, is generally 1–3 L/min, and can only be assessed by connecting the device to the bedside controller. Nominal operating speed on the HeartMate 3 is 4800–6500 rpm, drawing 3–6 watts with mean flow 3–6 L/min. The HeartMate 3 does not produce real time flow estimation waveforms like the HVAD. However, both the patient and bedside controllers display the pulsatility index—“PI”—a measure designed to indicate peak-trough flow pulsatility. PI is calculated by the device’s internal computer and is approximately equal to systolic minus diastolic flow divided by the mean. Typical PI values are 2–6, but some patients will have baselines outside of that range, especially on the higher end. The typical LVAD patient can often recall their typical pump parameters.

LVAD parameters that deviate from the patient or device baseline can be suggestive of problems, however pump parameters can never be interpreted without the context of a history and physical exam. As an illustrative example, hypovolemia will cause low flow and lower pulsatility /PI due to underfilling of the LV and decreased systolic contractility. Excessive hypovolemia, however, may lead to suction, evidenced by low-flow and higher pulsatility /PI due to intermittent flow stoppage when the myocardium occludes the inflow cannula. Hypervolemia, meanwhile, may lead to slightly higher mean flows with higher pulsatility /PI. However, excessive hypervolemia will lead to right ventricular failure, increased ventricular interdependence and septal leftward shift with compromise of LV contractility. This leads to decreased flow and pulsatility /PI (Table 1). Thus, history-taking and physical examination are paramount, even in the setting of an advanced device such as an LVAD.

## 7. Conclusions

Although not without complications as detailed in this review, LVADs add quality and quantity of life for end-stage HF patients with limited options for medical therapy or transplantation. Survival after continuous-flow LVADs continues to improve, with one-year survival post-LVAD now comparable to heart transplantation and longer-term LVAD outcomes ever improving [55]. With a growing population on long-term mechanical circulatory support, it is crucial for non-mechanical circulatory support providers to be familiarized with initial assessment and treatment of LVAD-supported patients. In all cases, coordination with an interdisciplinary LVAD team is necessary to support our patients with the care standards they deserve.

## Figures and Tables

**Figure 1 jcm-11-02575-f001:**
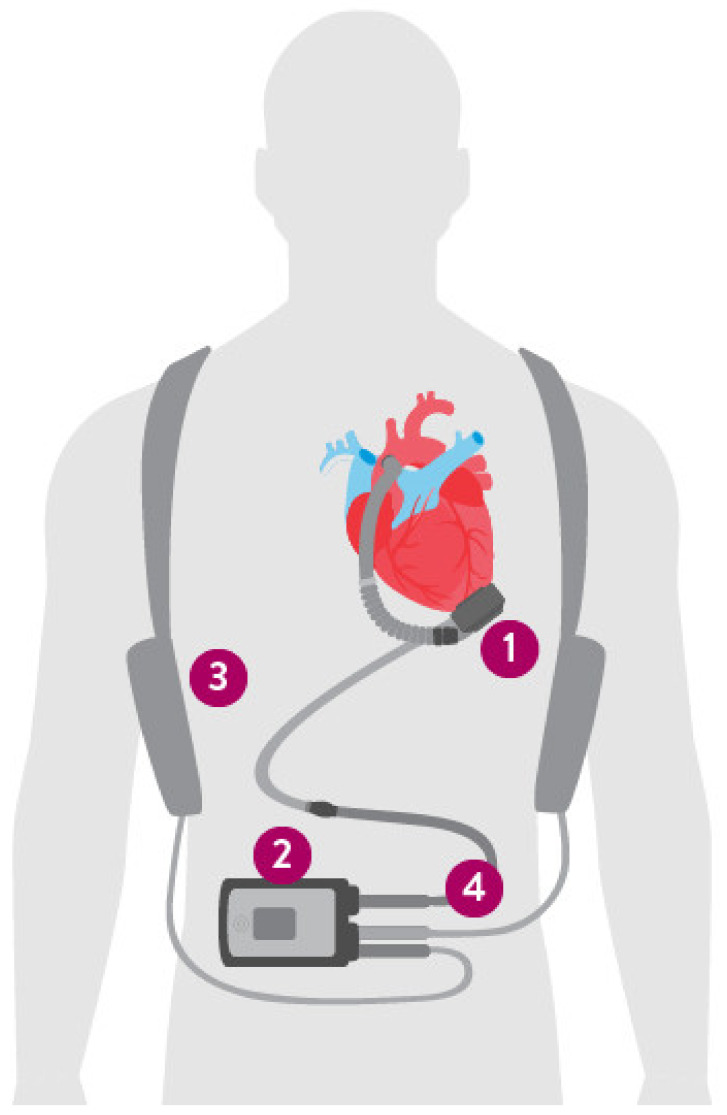
Contemporary LVAD device. Left ventricular assist device (LVAD) with full magnetic levitation with patient peripherals shown. The rotor pump is implanted on the left ventricular apex with outflow graft directed to the aortic root (**1**). A controller unit is typically worn on the belt and displays device settings, status, and alerts/alarms (**2**). The controller is connected to two portable battery packs (**3**), shown here suspended on the shoulders. The controller is attached to the LVAD via a driveline (**4**) that typically exits the skin in the upper abdomen and is covered with a sterile dressing at all times. (Image reprinted with permission from Abbott Laboratories).

**Figure 2 jcm-11-02575-f002:**
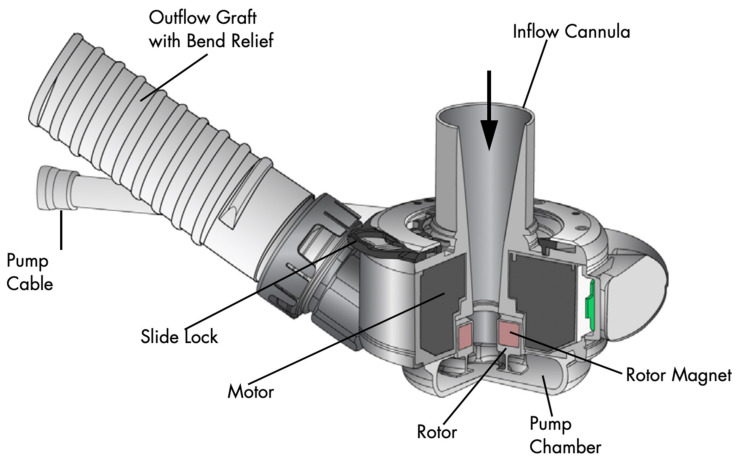
Device mechanics, detail. A HeartMate 3 left ventricular assist device (LVAD) shown with device mechanics. The inflow cannula is implanted into the left ventricular apex with the opening directed toward the mitral valve. Blood is circulated through the pump chamber via a spinning, magnetically levitated rotor which nominally operates at 5000–6000 rpm. Centrifugal effect forces the blood into the outflow graft, which meets the ascending aorta at a surgical anastomosis. (Image reprinted with permission from Abbott Laboratories).

**Figure 3 jcm-11-02575-f003:**
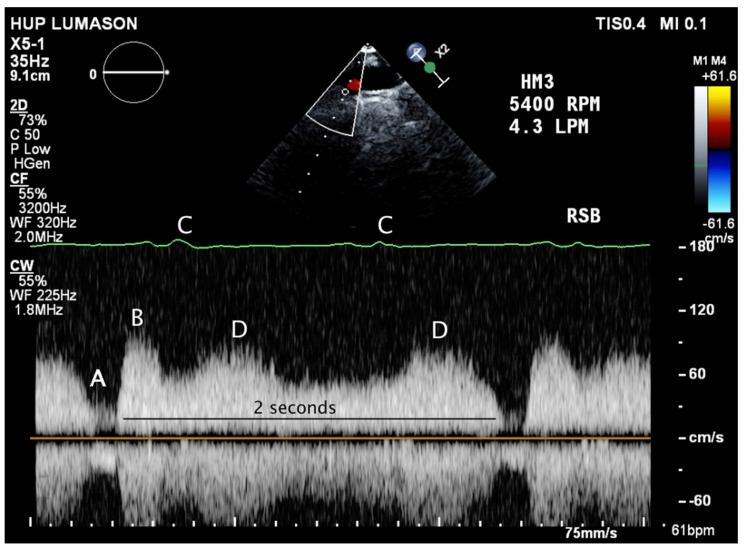
Echocardiography of LVAD patient. Continuous wave Doppler interrogation of the aorta from the right sternal border demonstrating the unique physiology of a patient with a normally functioning HeartMate 3 left ventricular assist device (LVAD). Every 2 s, the artificial pulse feature causes an abrupt decline (**A**), then increase (**B**) in flow delivered to the aorta. Following electrical systole on the ECG tracing (**C**), mechanical ventricular systole decreases the aorta-ventricular pressure gradient, which increases flow through the pump (**D**). An analogous tracing would be found on an invasive arterial line in the critical care unit. Despite the flow variation during the cardiac and LVAD “artificial pulse” 30 Hz cycle, this patient does not have a palpable pulse and the aortic valve does not open.

**Figure 4 jcm-11-02575-f004:**
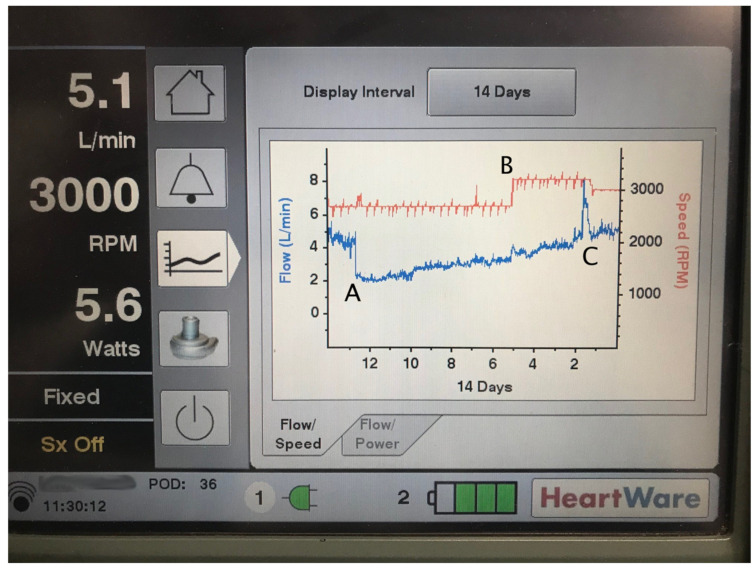
Device thrombus. A controller console showing an illustrative example of left ventricular assist device (LVAD) thrombosis in an HVAD system. Note the time scale of 14 days on the x-axis. This patient developed acute thrombosis at the inflow cannula 13 days ago, leading to dramatically increased afterload (increased effective gradient between left ventricle and aorta) and decreased LVAD flow (**A**). The thrombosis partially broke up over the next 7 days. Five days ago, the pump speed was increased with minimal effect on flow (**B**). Two days ago, the thrombus suddenly dislodged and was ingested by the rotor, causing a “power spike” (**C**), followed by return of baseline pump function. This patient tolerated these events well and went on to receive a heart transplant.

**Table 1 jcm-11-02575-t001:** Parameters in selected physiologic states for centrifugal-flow LVADs.

Condition	Flow	Pulsatility/PI	MAP	Comments
Hypovolumia				Decreased LV filling leads to decreased LV contractility and lower pulsatility. Extreme hypovolemia leads to suction, and sudden increase in pulsatility as flow intermittently drops to zero during suck-down.
mild	**⇔**/**⇓**	**⇓**	**⇔**	
severe (with suction)	**⇓**	**⇑⇑**	**⇔**/**⇓**	
Hypervolemia				Increased LV end diastolic pressure leads to increased LV contractility and pulsatility. Unchecked, however, the patient may progress to RV failure, increased LV-RV interdependence, impaired LV contractility and blood return.
mild	**⇔**/**⇑**	**⇑**	**⇔**	
severe (with RV failure)	**⇓**	**⇓**	**⇔**/**⇓**	
Hypertension	**⇓**	**⇑**	**⇑**	Increased afterload leads to substantial decrease in diastolic >> systolic flow, leading to larger pulsatility/PI.
Inflow/outflow obstruction	**⇓**	**⇑**	**⇔**/**⇓**	Creates a high afterload condition analogous to hypertension.
Rotor thrombus	“**⇑⇑**”	**⇔**/**⇓**	**⇔**/**⇓**	Increased power consumption from rotor weighted down by thrombus causes “power spikes” and high flow/power alarms. High flow is an artifact—effective flow is low.

Pulsatility refers to variation in flow through the pump between systole (peak) and diastole (trough), but not necessarily patient pulse pressure or palpable pulse on physical examination. LV = left ventricle, LVAD = left ventricular assist device, MAP = mean arterial pressure, PI = pulsatility index, RV = right ventricle. Key: **⇑** increased, **⇓** decreased, **⇔** no change.

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
