# Peer review of "Left Ventricular Assist Devices: A Primer for the Non-Mechanical Circulatory Support Provider"

_jcm, 2022, doi:10.3390/jcm11092575_

Round 1

Reviewer 1 Report

This is a kind of review article about LVAD therapy in the north America and western European country. The situation around LVAD therapy has been progressing rapidly. It is important to summarize the history of MCS after Novacor appeared. This article is apparently not the one for the experts of MCS therapy but for doctors of other field and health insurance business personnel. Nothing is wrong with the description and the description here is well organized and correct. This article will serve very well as an introductory article.  Non-MCS providers will be benefitted when they encounter patients with LVAD in the situation of daily practice.

According to the reason I mentioned above, there is no scientifically significant value in this article but may help "the primers". 

For me as a non-English native speaker, some expressions were difficult to follow at the beginning (device therapy short of MCS, mnemonic, etc). 

Reviewer 2 Report

  • Please describe in the introduction the other indications for LVAD implantation as “bridge to recovery” or “bridge to candidacy"

  • Please explain better the advantages of continuous axial flow instead of pulsatile or centrifugal flow pump in "Toward the Modern Continuous Flow Era"

  • In the section on “Thrombosis”, it may be useful to define the treatment therapy, the role of thrombolysis, and the possible need for an urgent transplant in the case of bridge-to-transplant indication

  • In the paragraph “Right ventricular failure”, please underline the need for a correct right ventricular function as a condition to implant  an LVAD

Reviewer 3 Report

The authors have developed a thorough review for non-HF & MCS specialists in appreciating the history, intricacies, and evaluation of LVAD. The flow of the manuscript allows the healthcare professional to appreciate the long journey in current LVADs, especially as a destination therapy. They then highlight the most common problems that are encountered, with a basic management approach. The algorithms and table are very helpful in summarizing the information. It is well organized with the material appropriate for healthcare professionals. 
